# Gas-responsive porous magnet distinguishes the electron spin of molecular oxygen

Wataru Kosaka [1,2], Zhaoyuan Liu[2], Jun Zhang[2], Yohei Sato[3], Akihiro Hori[4], Ryotaro Matsuda[4], Susumu Kitagawa [3] & Hitoshi Miyasaka [1,2]

Gas-sensing materials are becoming increasingly important in our society, requiring high sensitivity to differentiate similar gases like $N_2$ and $O_2$. For the design of such materials, the driving force of electronic host-guest interaction or host-framework changes during the sorption process has commonly been considered necessary; however, this work demonstrates the use of the magnetic characteristics intrinsic to the guest molecules for distinguishing between diamagnetic $N_2$ and $CO_2$ gases from paramagnetic $O_2$ gas. While the uptake of $N_2$ and $CO_2$ leads to an increase in $T_C$ through ferrimagnetic behavior, the uptake of $O_2$ results in an $O_2$ pressure-dependent continuous phase change from a ferrimagnet to an antiferromagnet, eventually leading to a novel ferrimagnet with aligned $O_2$ spins following application of a magnetic field. This chameleonic material, the first with switchable magnetism that can discriminate between similarly sized $N_2$ and $O_2$ gases, provides wide scope for new gas-responsive porous magnets.

[1] Institute for Materials Research, Tohoku University, 2-1-1 Katahira, Aoba-ku, Sendai 980-8577, Japan. [2] Department of Chemistry, Graduate School of Science, Tohoku University, 6-3 Aramaki-Aza-Aoba, Aoba-ku, Sendai 980-8578, Japan. [3] Institute for Integrated Cell-Materials Science (iCeMS), Kyoto University, Katsura, Nishikyo-ku, Kyoto 615-8510, Japan. [4] Department of Materials Chemistry, Graduate School of Engineering, Nagoya University, Furo-cho, Chikusa-ku, Nagoya 464-8603, Japan. Correspondence and requests for materials should be addressed to H.M. (email: miyasaka@imr.tohoku.ac.jp)

I n this Internet of Things age[1], it is essential to control how information is processed when only slight differences within the data exist, leading to the notion of sensing. The development of highly sensitive devices for ubiquitous gas and innocuous small molecule sensing is one of the major challenges in the field of materials science[2]. A magnetic change can be beneficial for providing a responsive signal in such a sensing device, and would be advantageous for gas detection owing to contactless operation and detection independent of the sample shape of the host framework. Further, devices that respond quickly with easy operability and readability for ON/OFF updates are desirable; the availability of spin freedom in host-guest interactions for gas sensing is an innovative technique that could make this possible. For instance, distinguishing between nitrogen ($N_2$) and oxygen ($O_2$) gases is exceedingly difficult because of their similar size and boiling points[3,4]. Detecting a magnetic change induced by the intrinsic magnetic nature of these gases (i.e., diamagnetic $N_2$ and paramagnetic $O_2$) would represent a major breakthrough in gas-sensing technologies. For this purpose, however, a drastic phase change in magnetism, not just small modifications of magnetic properties[5,6], is necessary. The gas-induced magnetic response has also been investigated using $Fe^{II}$ spin-crossover systems;[7–9] however, magnetic discrimination between $O_2$ and $N_2$ has never been observed. Meanwhile, drastic magnetic changes induced by solvation/desolvation[10] have prompted lively discussions on magnetic sponges[11–17] and spin-crossover systems[18–21]. Despite this, a strong magnetic response to gases in air such as $N_2$, $O_2$, and carbon dioxide ($CO_2$), which possess relatively small sizes and low or no reactivity and electric polarity, remains a significant challenge for the development of functional porous magnetic materials.

Here, we report a porous layered ferrimagnet that reversibly alters its magnetic phase in response to the magnetic type of the inserted gas, i.e., diamagnetic for $N_2$ and $CO_2$ or paramagnetic for $O_2$. The fully $O_2$-adsorbed compound changes to an antiferromagnet, but application of a magnetic field results in a unique ferrimagnetic phase where some of the oxygen spins become aligned synergistically. Recently, the control of spin coupling on oxygen molecules inserted into molecular porous frameworks[22–27] or graphite[28,29], as well as in bulk materials[30–33], has been seen as an important topic. Nevertheless, this is the first case in which a paramagnetic phase resulting from condensed oxygen molecules plays a key role for long-range ordering in an $O_2$-accommodated magnet.

## Results

**Crystal structure of the pristine framework**. To develop gas-responsive porous magnets, we chose a layered ferrimagnet, $[\{Ru_2(3,5\text{-}F_2PhCO_2)_4\}_2\{TCNQ(MeO)_2\}]\cdot3(DCM)\cdot1.5(DCE)$ (**1-solv**; $3,5\text{-}F_2PhCO_2^-$ = 3,5-difluorobenzoate; $TCNQ(MeO)_2$ = 2,5-dimethoxy-7,7,8,8-tetracyanoquinodimethane; DCM = dichloromethane; DCE = 1,2-dichloroethane), obtained from an electron-donor (D)/-acceptor (A) 2:1 assembly that involves an electron transfer[15,34–40], where the paddlewheel-type $[Ru_2(3,5\text{-}F_2PhCO_2)_4]$ subunit (abbreviated as $[Ru_2]$) is D and $TCNQ(MeO)_2$ is A. Compound **1-solv** crystallized in the triclinic space group $P-1$, where two different $[Ru_2]$ units and one $TCNQ(MeO)_2$ molecule, with respective inversion centers, were structurally isolated ($Z = 1$) with a charge assignment of $[-\{Ru(1)_2^{II,III}\}^+-\mu_4\text{-}TCNQ(MeO)_2^{\bullet-}-\{Ru(2)_2^{II,II}\}-]$ (Fig. 1a,b, Supplementary Fig. 1a, Supplementary Table 1–3, Supplementary Note 1 and 2). The set of two $[Ru_2]$ units and $TCNQ(MeO)_2$ constructs a fishnet-like two-dimensional network lying on the (100) plane that stacks along the $a$-axis (Fig. 1a,b). The inter-layer distances defined by the vertical ($l_1$) and inter-unit translational ($l_2 = a$-

axis; Fig. 1b) distances between the planes are 9.78 Å and 10.65 Å, respectively (Supplementary Table 8), and the crystallization solvents (3(DCM)·1.5(DCE)) are located between the layers with a solvent accessible volume of 713 Å$^3$ (32% of total volume).

**Crystal structure of the dried phase**. Compound **1-solv** gradually releases the crystallization solvents upon increasing temperature, producing the solvent-free porous compound $[\{Ru_2(3,5\text{-}F_2PhCO_2)_4\}_2\{TCNQ(MeO)_2\}]$ (**1**), which is stable at temperatures up to 450 K with its crystallinity intact (Supplementary Fig. 2a). Similar to **1-solv**, **1** crystallized in the triclinic $P-1$ space group ($Z = 1$, Fig. 1c,d, Supplementary Fig. 1b, Supplementary Table 1 −3, Supplementary Note 1 and 2). Although the fishnet-like network was preserved with a slightly shortened (relative to **1-solv**) $l_1$ of 9.46 Å, while with a lengthened $l_2$ of 10.84 Å ( $= a'$-axis; Fig. 1d), the structural features of the network are drastically altered from an almost flat form in **1-solv** to a wavy form in **1** (Fig. 1c,d,e), resulting in a reduction of the void volume to 147 Å$^3$ (7.5% against total volume). Of note, the electronic state of $[Ru_2]$ and $TCNQ(MeO)_2$ units in **1** remains unchanged (Supplementary Table 2, 3, Supplementary Note 1). Compound **1** becomes **1-solv** when exposed to DCM/DCE vapor for 12 h at 300 K (Supplementary Fig. 2b), indicating a common magnetic sponge behavior (vide infra).

**Magnetic sponge behavior**. The spins of the $[Ru_2^{II,II}]$ ($S = 1$) and $[Ru_2^{II,III}]^+$ ($S = 3/2$) moieties interact antiferromagnetically with the radical spin of $TCNQ(MeO)_2^{\bullet-}$[41,42] over the layered network forming a ferrimagnetically ordered layer, which is followed by three-dimensional ferrimagnetic ordering with inter-layer ferromagnetic interactions[15,34,36,38–40]. The magnetic transition temperature $T_C$ (or $T_N$ for antiferromagnetic ordering) for this type of layered magnetic material should be strongly affected by intra-layer exchange interactions between the $[Ru_2]^{0/+}$ units and $TCNQ(MeO)_2^{\bullet-}$, as well as inter-layer dipole interactions[15,36,38–40,43]. Figure 1f shows the temperature dependence of field-cooled dc magnetization (FCM) of **1-solv** and **1** in a 1 kOe dc field ($H_{dc}$). In both compounds, an abrupt increase in the FCM is observed near 80 K without a subsequent decrease at lower temperatures. This occurs independent of the applied fields, indicating the onset of ferrimagnetic ordering[35,37] (details of the comparison between **1-solv** and **1** are described in Supplementary Fig. 3 and Supplementary Note 3 and 4); however, their $T_C$ values differ (i.e., 83 K and 76 K for **1-solv** and **1**, respectively), as evaluated from remnant magnetization (RM) (inset of Fig. 1f) and ac susceptibility data (Supplementary Fig. 3b, e, Supplementary Note 3).

**Gas sorption capability**. In addition to the magnetic sponge capabilities for crystallization solvents, **1** has the ability to adsorb gases such as $CO_2$, $N_2$, and $O_2$; the gas-adsorbed phase is defined as **1 ⊃ Gas**. Figure 2a shows their sorption isotherms (a log-scale plot is shown in Supplementary Fig. 4). For $N_2$, **1** has a non-porous nature at 77 K because of the slow diffusion of gaseous molecules into the void space; however, **1** acts as an adsorbent at 120 K, where the 1st gate-opening is observed at a pressure of 3.2 kPa, as found in other low-dimensional porous systems[44], and reaches an adsorption amount of 27 mL (stp) g$^{-1}$ (2.3 mol per formula unit) at 99 kPa. The $CO_2$ adsorption isotherm at 195 K shows a steep rise at relatively low pressures, where the adsorption amount is 102 mL (stp) g$^{-1}$ (8.7 mol per formula unit) at 99 kPa[45], even though a gate-opening modification should be involved. The $O_2$ adsorption isotherm at 90 K shows a stepwise feature; 1st and 2nd gate-opening transitions at ca. 0.1 kPa and 36 kPa, respectively, reaching an adsorbed amount of 110 mL (stp) g$^{-1}$ (9.5 mol per formula unit) at 99 kPa; however, only the

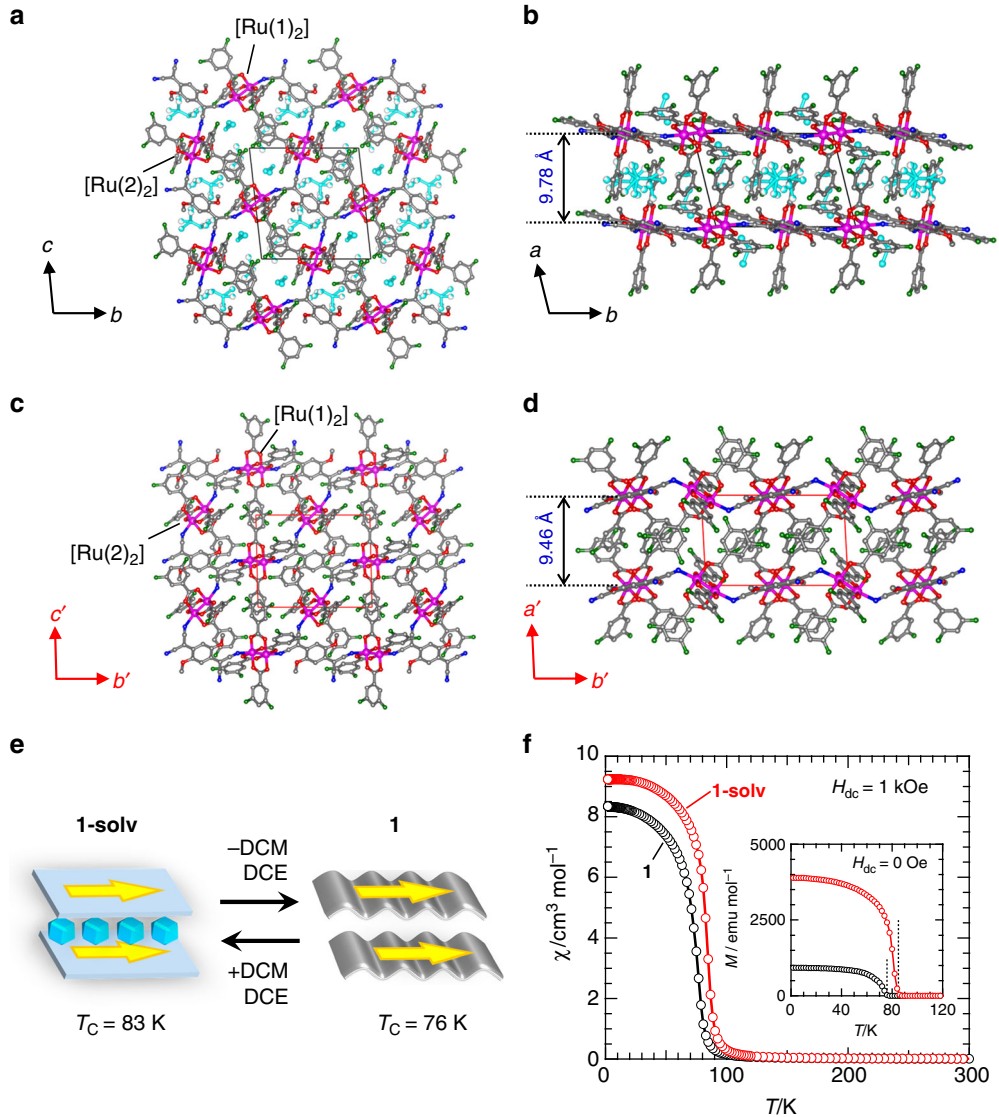

**Fig. 1** Structural modulation and magnetic sponge behavior upon solvation/desolation. **a–d** Views of the crystal structures of **1-solv** (**a**, **b**) and **1** (**c**, **d**); figures **a**, **c** and **b**, **d** show projections along the *a* (*a'*) axis and *c* (*c'*) axis, respectively, where atoms N, O, C, F, and Ru are represented in blue, red, gray, green, and purple, respectively, and the crystallization solvents given in figures **a** and **b** are represented in cyan. The vertical inter-layer distance ($l_1$) is indicated in blue digit in **b** and **d**, and the inter-unit translational distance ($l_2$) corresponds to the *a*- or *a'*-lattice axis for each compound (Supplementary Table 8). The cell axes *a'*–*c'* in **c** and **d** represent those for a transformed lattice (see Supplementary Methods), which were adopted for easy comparison with the lattice of **1-solv**. **e** Schematic representation for the interchange between **1-solv** and **1** upon solvation/desolvation, where the colored arrow indicates a plausible direction of ordered spins. **f** Temperature dependence of magnetic susceptibility ($\chi$) for **1-solv** (red) and **1** (black) measured under a 1 kOe dc field ($H_{dc}$) on field cooled process. Inset: Remnant magnetization ($M$) at $H_{dc} = 0$ Oe (heating process) measured after taking a FCM (1.8–120 K) under a 3 Oe dc field, where the dashed lines indicate the respective $T_C$ values

1st gate-opening at ca. 3.1 kPa is observed when measured at 120 K, eventually reaching an adsorbed $O_2$ amount of 64 mL (stp) $g^{-1}$ (5.5 mol per formula unit) at 99 kPa.

**Crystal structures under gases**. To elucidate the gas-inserted structure, in situ powder X-ray diffraction (PXRD) of **1** were measured under 100 kPa of $N_2$ at 130 K, $O_2$ at 94 and 130 K, and $CO_2$ at 204 K (Fig. 2b), which illustrate the occurrence of structural transformations upon gas adsorption. Two types of gas-adsorbed temperature-dependent phases exist at 130 and 94 K under $O_2$, which can be associated with the 2nd gate-opening step in the adsorption isotherm for $O_2$. Additionally, the PXRD pattern of **1 ⊃ O₂** at 130 K is very similar to that of **1 ⊃ N₂** at 130 K in that it does not undergo the 2nd gate-opening transition.

Hereafter, $O_2$-adsorbed phases observed at 130 K and 94 K are denoted as **1 ⊃ O₂-I** and **1 ⊃ O₂-II**, respectively. Notably, the gas-induced structural changes are reversible (Supplementary Fig. 5); after evacuating the $CO_2$ gas from **1 ⊃ CO₂**, the PXRD pattern reverts to the original pattern of **1**. In the case of **1 ⊃ N₂**, slight heating to 150 K in addition to evacuation is required to promote desorption of $N_2$. Of note, the PXRD pattern for **1 ⊃ O₂-II** becomes that for **1 ⊃ O₂-I** by evacuating at 94 K, but it does not return to the pattern of **1**, indicating that the **1 ⊃ O₂-I** phase corresponds to an intermediate phase stabilized at low pressures of $O_2$ even at 94 K (vide infra), which eventually turns into **1** after evacuating at 300 K.

Finally, the crystal structures of **1 ⊃ N₂**, **1 ⊃ O₂-I**, and **1 ⊃ CO₂** were determined by in situ single crystal X-ray diffraction (SCXRD) under gas-pressure controlled atmospheres

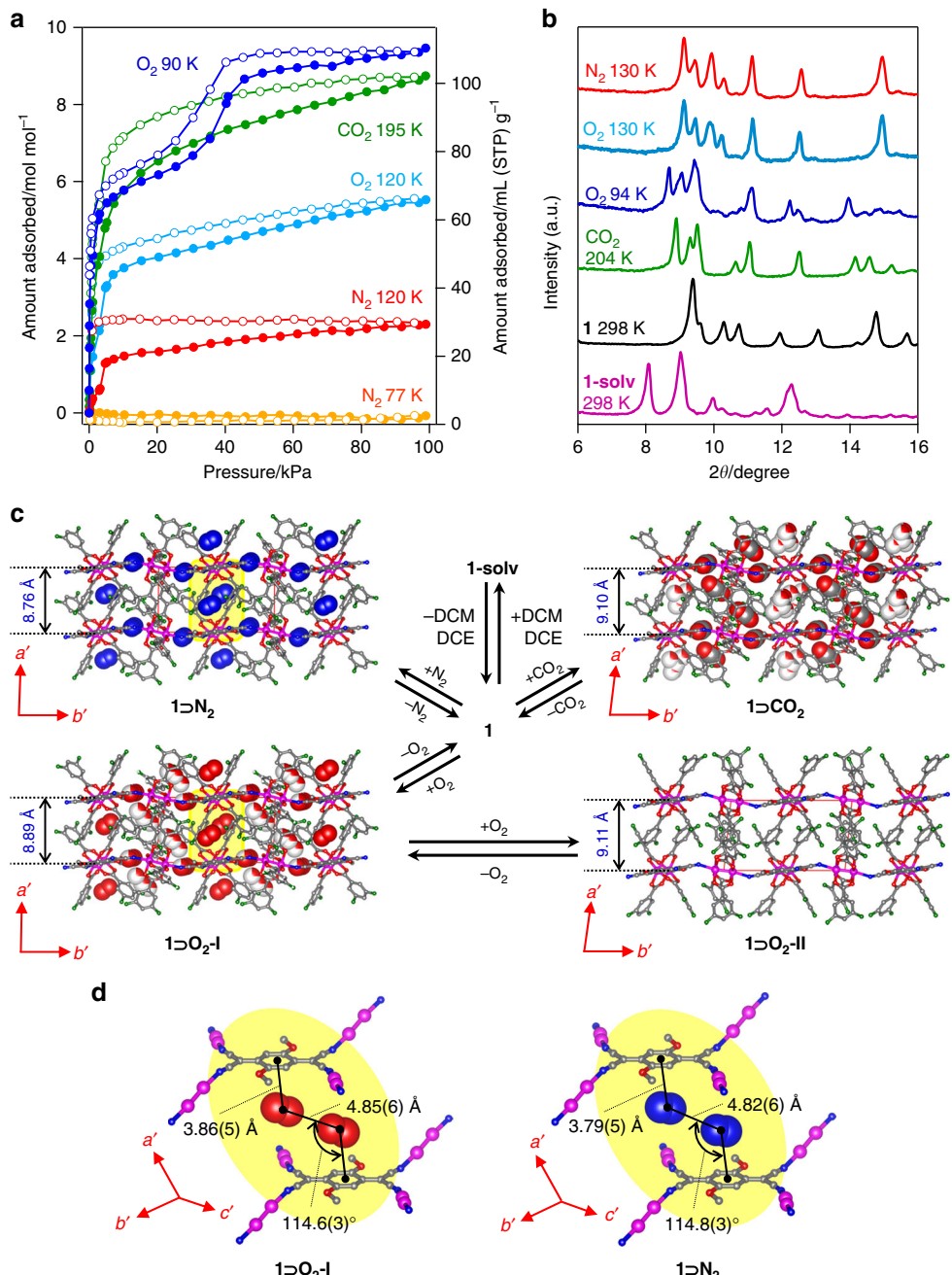

**Fig. 2** Gas sorption capability, sorption-induced structural transitions, and crystal structures under gases. **a** Adsorption (closed) and desorption (open) isotherms of **1** for $CO_2$ at 195 K (green), $N_2$ at 77 K (orange) and 120 K (red), $O_2$ at 90 K (blue) and 120 K (cyan), where solid lines are only guide for the eye. **b** PXRD patterns measured ($\lambda = 1.54$ Å) under 100 kPa of $N_2$ at 130 K (red), $O_2$ at 130 K (cyan) and 94 K (blue), and $CO_2$ at 204 K (green) and ones for **1** (black) and **1-solv** (purple) at 298 K. **c** Packing views ($c'$-axis projection, where atoms N, O, C, F, and Ru are represented in blue red, gray, green, and purple, respectively) of $\mathbf{1 \supset N_2}$ at 130 K, (upper left), $\mathbf{1 \supset O_2\text{-}I}$ at 130 K (lower left), $\mathbf{1 \supset O_2\text{-}II}$ at 94 K (lower right), and $\mathbf{1 \supset CO_2}$ at 195 K (upper right) and their transformation routes including **1-solv** and **1**, where the structures for $\mathbf{1 \supset N_2}$, $\mathbf{1 \supset O_2\text{-}I}$, and $\mathbf{1 \supset CO_2}$ were determined by in situ SCXRD analyses, and it for $\mathbf{1 \supset O_2\text{-}II}$ was evaluated from the PXRD pattern with the Rietveld refinement (the accommodated gas molecules in a space filling model, of which the site occupancy is colored, are only represented in $\mathbf{1 \supset N_2}$, $\mathbf{1 \supset O_2\text{-}I}$, and $\mathbf{1 \supset CO_2}$, and minor position-disordered components in a part of layer framework in $\mathbf{1 \supset O_2\text{-}I}$ were omitted for clarity). The $a'$–$c'$ axes represent those of a transformed lattice for easy comparison with the lattice of **1-solv** (see Supplementary Methods). The vertical inter-layer distances ($l_1$) are indicated in blue digit, and the inter-unit translational distance ($l_2$) corresponds to the $a'$-lattice axis for each compound (Supplementary Table 8). **d** A close-up view of gas-sandwiched mode at Site-A (Supplementary Fig. 10) denoted as an yellow ellipse in figure c for $\mathbf{1 \supset O_2\text{-}I}$ and $\mathbf{1 \supset N_2}$

(Supplementary Fig. 6, 7, 9, and 10, Supplementary Table 4−6, Supplementary Note 5−7), and it of $\mathbf{1 \supset O_2\text{-}II}$ was evaluated from the in situ PXRD data using the Rietveld refinement technique (Supplementary Fig. 8, Supplementary Table 7, Supplementary Note 6). For $\mathbf{1 \supset N_2}$, $\mathbf{1 \supset O_2\text{-}I}$, and $\mathbf{1 \supset CO_2}$, the

accommodated gases were reasonably determined with occupancy numbers of $4N_2$, $5.2O_2$, and $5CO_2$, respectively, which were displayed in Fig. 2c and Supplementary Fig. 9 and 10 (the $O_2$ molecules for $\mathbf{1 \supset O_2\text{-}II}$ have less accuracy, so only the framework structure is discussed). The inter-layer distances, which are

defined by $l_1$ and $l_2$ ( $= a'$-axis) between planes, have decreased (but $l_2 > 10.3$ Å; Supplementary Table 8, where inter-layer ferromagnetic interaction is expected even for $1 \supset O_2$-I and $1 \supset O_2$-II)[39,40] in all $1 \supset$ **Gas** structures compared to **1**, which manifests structurally as a change from the wavy layer form in **1** to a quasi-flat layer form in $1 \supset$ **Gas**, resulting from a reduction in the Ru−N−C bending angle (av. 159–161° for $1 \supset$ **Gas** vs. 140.3° for **1**, Supplementary Table 8). Hence, the guest-molecule accessible volume between the layers in $1 \supset$ **Gas** increase relative to **1** (147 Å$^3$), as expected from the gas adsorption capability (335–546 Å$^3$ for $1 \supset$ **Gas**, Supplementary Table 8). Importantly, the structural frameworks of $1 \supset N_2$ and $1 \supset O_2$-I at 130 K are almost identical; three distinguishable gas-accommodation sites (Site-A–C; Supplementary Fig. 10) were commonly realized even for $1 \supset CO_2$, where a close-up view of gas-sandwiched mode at Site-A, which was most likely associated with the difference of magnetism between $1 \supset N_2$ and $1 \supset O_2$-I (vide infra), was depicted in Fig. 2d. The Site-A included two molecules of $N_2$ or $O_2$ with a similar arrangement; two gas molecules at Site-A were relatively close to the $TCNQ(MeO)_2$ moieties, where the barycenter-to-barycenter distance between $N_2/O_2$ and the quinonoid ring (C6) of the $TCNQ(MeO)_2$ moiety was 3.79(5) and 3.86(5) Å, respectively, and the inter-guest barycenter distance of $N_2{\cdots}N_2/O_2{\cdots}O_2$ was 4.82(6) and 4.85(6) Å, respectively (Fig. 2d). The torsion angle of $C6{\cdots}N_2/O_2{\cdots}N_2/O_2$ was 114.8(3)° and 114.6(3)° for $1 \supset N_2$ and $1 \supset O_2$-I, respectively (Fig. 2d). To accommodate an additional 4–8 mol per formula unit of gas, a subsequent enlargement in the inter-layer distance is required, as observed in $1 \supset O_2$-II and $1 \supset CO_2$ (Fig. 2c).

**Magnetic properties under diamagnetic gases, $CO_2$ and $N_2$.** Upon gas adsorption, a significant structural change is induced without alteration in the oxidation state of each unit in the $D_2A$ layer; in situ infrared (IR) spectroscopy proves the preservation of $TCNQ(MeO)_2^{\bullet-}$, even under a 100 kPa gas atmosphere (Supplementary Fig. 11). Therefore, in situ magnetic measurements were conducted in Quantum Design MPMS-7S by accurately handling the gas pressure; the pressure in a homemade cell (Supplementary

Fig. 12) containing the sample was evacuated down to 0.1 Pa with a turbo-molecular pump at 353 K and the gas was introduced at 200 K up to an inner gas pressure of ~116 kPa. The gas-sealed cell was then cooled at a sweep rate of 0.5 K min$^{-1}$ to 120 K for $N_2$, 195 K for $CO_2$, and 100 K for $O_2$. Each cell was maintained at its respective temperature for 10 h to reach adsorption equilibrium. Once the inner pressure of each cell was obtained, the gas-sealed cell was held at 100 K or 120 K for the FCM measurements.

Figure 3a shows the temperature dependence of FCM at 100 Oe for $1 \supset N_2$ and $1 \supset CO_2$ prepared in situ, together with that for **1**. Upon insertion of $N_2$ and $CO_2$, $T_C$ drastically increases to 88 K for $1 \supset N_2$ and 92 K for $1 \supset CO_2$ from 76 K for **1** (under vacuum) even under a weaker magnetic field of 5 Oe (Supplementary Fig. 13), establishing the existence of a ferrimagnetic ground state under $N_2$ and $CO_2$ atmospheres, where $T_C$ was determined from a disappeared point of RM (Supplementary Fig. 14). Since $N_2$ exists in the gas phase at 88 K in bulk, the change in $T_C$ is not caused by external $N_2$. In addition, $N_2$ and $CO_2$ are diamagnetic species. Therefore, the variation in $T_C$ results from the adsorbed gases. Given that the decrease in $T_C$ from **1-solv** to **1** was induced by considerable structural changes, inversely, the increase in $T_C$ for $1 \supset N_2$ and $1 \supset CO_2$ relative to **1** likely results from a reduction in structural deformation; the wavy layer in **1** is modified into a quasi-flat layer in $1 \supset N_2$ and $1 \supset CO_2$ and/or a modification in the inter-layer environment occurs, resulting from closely packed gases (Fig. 3c). Even with such a drastic change in $T_C$, the magnetic-field dependence of the magnetization ($M$–$H$) is essentially preserved from **1** (Fig. 3b), although the coercive field ($H_c$) of $1 \supset CO_2$ is somewhat larger than that of **1** and $1 \supset N_2$. Note that the anomalous steps around zero field for **1**, $1 \supset N_2$, and $1 \supset CO_2$ ($1 \supset O_2$ as well; vide infra) could be caused by a small number of free crystals that follow the magnetic field.

**Magnetic properties under a paramagnetic $O_2$ gas.** The magnetic behavior of the material under an $O_2$ atmosphere is completely different from that under $N_2$ and $CO_2$ and varies with the $O_2$ pressure ($P_{O2}$) (Fig. 4). Similar to $1 \supset N_2$ and $1 \supset CO_2$, the

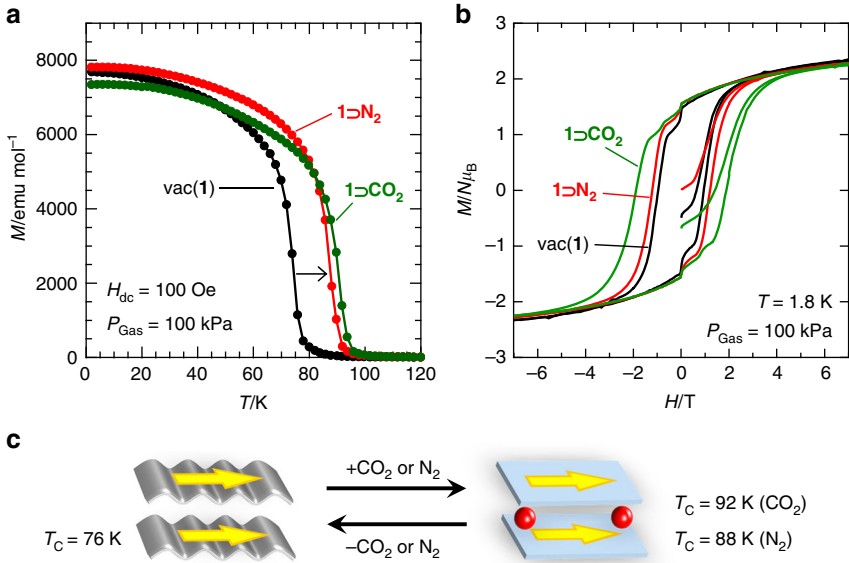

**Fig. 3** Variation of magnetic properties under diamagnetic gases, $CO_2$ and $N_2$. **a** FCM curves at a 100 Oe magnetic field for **1** measured under vacuum (black) and $1 \supset CO_2$ (green) and $1 \supset N_2$ (red) under a 100 kPa gas atmosphere. **b** Magnetic hysteresis loops at 1.8 K for **1** measured under vacuum (black), $1 \supset CO_2$ (green), and $1 \supset N_2$ (red) under a 100 kPa gas atmosphere. **c** Schematic representation for the alternation between **1** and $1 \supset CO_2$ and $1 \supset N_2$ upon $CO_2$ and $N_2$ adsorptions, respectively

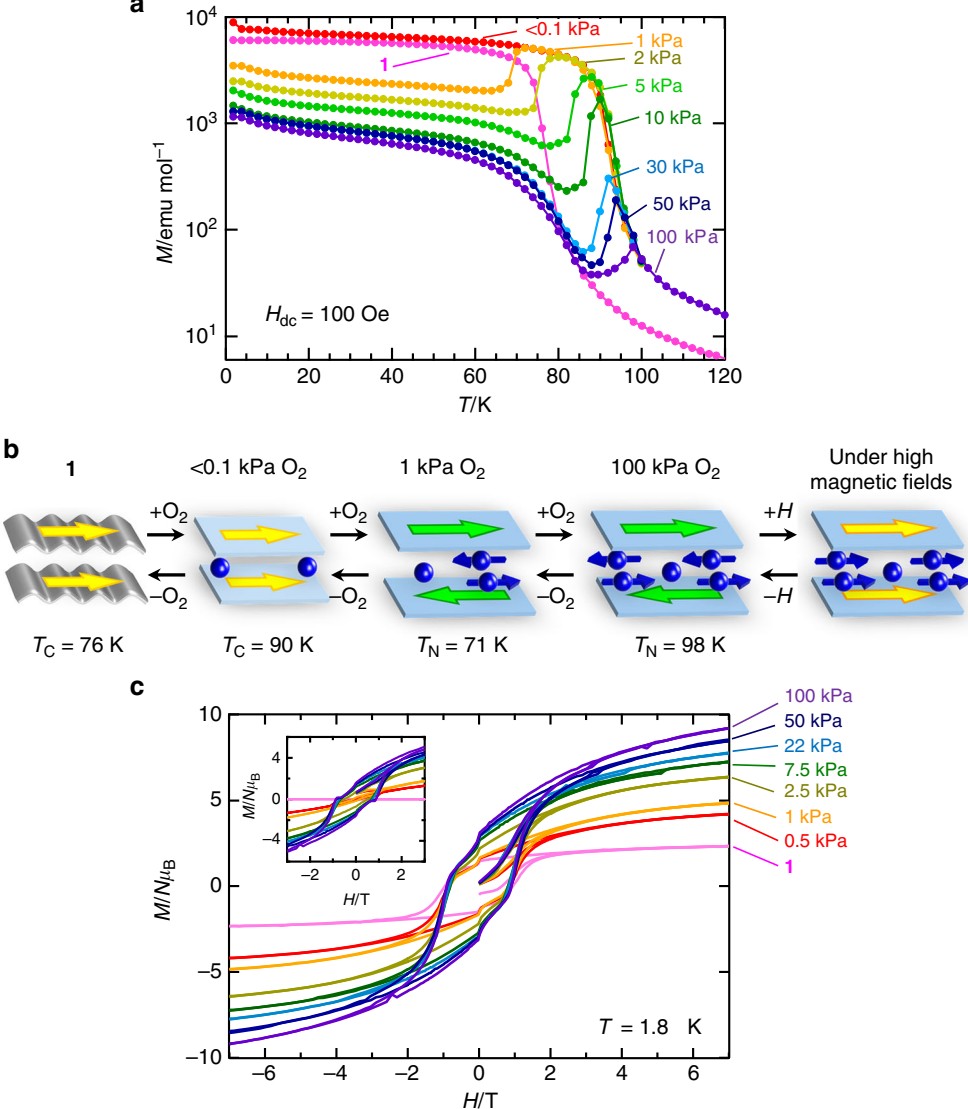

**Fig. 4** Variation of magnetic properties under a paramagnetic $O_2$ gas. **a** $O_2$-pressure dependence of FCM curves at a 100 Oe magnetic field for **1**. **b** Schematic representations for the alternations of **1 ⊃ $O_2$** dependent on the applied $O_2$ pressure and external magnetic fields. **c** Field-dependence of the magnetization at 1.8 K for **1** measured at several $O_2$ pressures, where the inset represents the differential plots on the basis of the $M-H$ curve for **1**

$T_C$ of **1 ⊃ $O_2$** increases once at low pressures of $P_{O2} < 1$ kPa (e.g., $T_C = 90$ K at $P_{O2} \leq 0.1$ kPa; vacuum pressure level at 100 K). However, under higher pressures, the FCM curve shows an anomaly with a cusp, indicating the onset of antiferromagnetic ordering; for example, $T_N = 71$ K at 1 kPa, which gradually increases to $T_N = 98$ K at 100 kPa with increasing $O_2$ pressure (Fig. 4a). The variation in $T_N$ with $O_2$ pressure was also confirmed by the magnetization measurements by varying the $O_2$ pressure at each temperature (Supplementary Fig. 15). The initial increase in $T_C$ at low $O_2$ pressures ($P_{O2} < 1$ kPa) is likely caused by the same mechanism found in **1 ⊃ $N_2$** and **1 ⊃ $CO_2$** (Fig. 3c), which could be attributed to the redress of the layered structure, i.e., the modification from a wavy form of **1** to a quasi-flat form in **1 ⊃ $O_2$-I** (the first step in Fig. 4b). Meanwhile, the drastic change of the magnetic phase from ferrimagnetism to antiferromagnetism could be obtained whether for: (1) a structural change associated with the transformation from **1 ⊃ $O_2$-I** to **1 ⊃ $O_2$-II**, or (2) the magnetic contribution of the adsorbed $O_2$ molecules. To examine these possibilities, PXRD patterns (from both of common lab level and high resolution synchrotron level) were

measured by varying the $O_2$ pressure at a fixed temperature in the range of 70–100 K (Supplementary Fig. 16 and 17), and the structural transition pressure ($P_c$) from **1 ⊃ $O_2$-I** to **1 ⊃ $O_2$-II** at each temperature was plotted in a $T–P_{O2}$ phase diagram together with $T_N$, where the $T_N$ line separates the magnetic phases between the paramagnetic/ferrimagnetic phase and the antiferromagnetic phase, and the $P_c$ line distinguishes between the **1 ⊃ $O_2$-I** and **1 ⊃ $O_2$-II** phases (Fig. 5). Importantly, the $T_N$ line is independent of the $P_c$ line, and antiferromagnetism in the **1 ⊃ $O_2$-I** phase is present (the pale blue area in Fig. 5). Since the **1 ⊃ $O_2$-I** and **1 ⊃ $N_2$** structures are identical with $l_2 > 10.3$ Å expected as a regime for inter-layer ferromagnetic interactions[39,40], and indeed, **1 ⊃ $N_2$** is ferrimagnetic, the antiferromagnetism in **1 ⊃ $O_2$-I** results from the magnetic contribution of the adsorbed $O_2$ molecules, which is caused by long-range antiferromagnetic correlations via intercalated $O_2$ spins; the most likely packing mode associated with the $O_2$-mediated magnetic pathway was shown in Fig. 2d. Further, the continuous shift in $T_N$ is likely dependent on the number of $O_2$ spins between layers, which act as magnetic mediators couple layer's ordered spins together

(Fig. 4b). Thus, the present porous layered magnet **1** magnetically discriminates $O_2$ from $N_2$ and $CO_2$, at least at $P_{O2} \geq 1$ kPa.

The magnetic switching between the ferrimagnetic phase under vacuum with the **1 ⊃ O₂-I** structure and the antiferromagnetic phase of **1 ⊃ O₂** is quite fast and reversible (Fig. 6); the change from the ferrimagnetic phase to the antiferromagnetic phase is completed in <1 min at 85 K.

Generally, the solid states of bulk $O_2$ exist in the $\alpha$-dimer form with a spin singlet at $T < 24\,K$[30,46]. Compound **1 ⊃ O₂-II** eventually has ~9 $O_2$ molecules per $D_2A$ layer unit, like a buried oxygen layer between ferrimagnetic $D_2A$ layers; at least, some of them certainly act as a paramagnetic mediator in the pores. Interestingly, the antiferromagnetic phase of **1 ⊃ O₂-II** transforms to a ferrimagnetic phase in the presence of an applied magnetic field (Supplementary Fig. 18), giving the much higher saturated magnetization ($M_s$) value of 9.29 $N\mu_B$ compared to 2.22

$N\mu_B$ for **1** at 7 T (1.8 K), including a fully opened hysteresis curve ($H_c = 0.70$ T) (Fig. 4c). On the basis of the $M–H$ curve for **1**, the differential plots clarify the contribution of the $O_2$ spins in the bulk magnetism of **1 ⊃ O₂-II** (Fig. 4d), giving rise to a new magnetic field-induced ferrimagnet. These magnetic alternations by gases are completely reversible upon adsorption/desorption under vacuum with heating (Supplementary Fig. 19).

## Discussion

The magnetic change caused by the introduction of guest gas molecules into a porous magnet can be attributed to three triggers: (i) an electronic trigger that causes spin emergence in the frameworks as a result of host-guest electron transfers (i.e., formation of new magnetic pathways in the framework); (ii) a structural trigger resulting from magnetostructural modifications associated with gate-opening-/-closing transitions induced by gas adsorption/desorption, respectively (i.e., modification of the magnetic pathways); and (iii) a paramagnetic guest trigger resulting from the formation of new magnetic pathways or dipole–dipole interactions where paramagnetic gas molecules themselves magnetically mediate the transition to another magnetic ground state. The present gas-responsive porous magnet results from triggers (ii) and (iii); in particular, the insertion of free oxygen molecules achieves a magnetic phase change from a ferrimagnet to an antiferromagnet based on trigger (iii). The fact of magnetic ordering via paramagnetic $O_2$ molecules gives an opportunity to investigate the intrinsic nature of oxygen molecules in closed nano-sized porous spheres and provides a new application methodology based on paramagnetic molecules as switchable magnetic mediators. As a rapidly emerging field, this class of gas-responsive porous magnets is the most important target in the development of functional molecular porous materials.

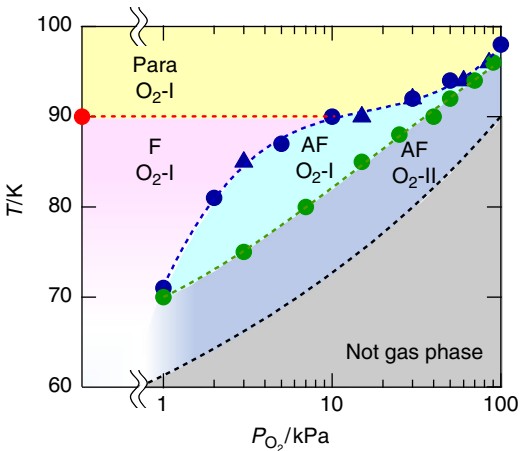

**Fig. 5** Structural and magnetic phase diagram for **1** under an $O_2$. Red and blue closed-circles represent $T_C$ and $T_N$ for ferrimagnetic and antiferromagnetic orderings, respectively, which were determined from the $M–T$ curves (Fig. 4a) measured under each $O_2$ pressure fixed at 100 K. Blue closed-triangles represent $T_N$ for antiferromagnetic ordering determined from the magnetization measurements by varying the $O_2$ pressure at each fixed temperature (Supplementary Fig. 15). Green closed-circles represent the structural transition pressure ($P_c$) from **1⊃O₂-I** to **1⊃O₂-II** determined by PXRD measurements by varying the $O_2$ pressure at each fixed temperature (Supplementary Fig. 16). Red dotted line separates magnetic phases between the paramagnetic (Para) phase and ferrimagnetic (F) phase. Blue dotted line ($T_N$ line) separates magnetic phases between the Para/F phase and the antiferromagnetic (AF) phase. The green dotted line ($P_c$ line) separates the **1⊃O₂-I** and **1⊃O₂-II** phases. Black dotted line represents the saturated vapor pressure curve, which distinguishes between the gas phase and non-gas phase for bulk $O_2$

## Methods

**Physical measurements.** IR spectra were measured with KBr pellets using a Jasco FT/IR-4200 spectrometer. Thermogravimetric analyses (TGA) were performed using a Shimadzu DTG-60H apparatus under a $N_2$ atmosphere in the temperature range from 298 K to 673 K at a heating rate of 5 K min$^{-1}$. Unless otherwise noted, PXRD were collected on a Rigaku Ultima IV diffractometer with Cu-$K\alpha$ radiation ($\lambda = 1.5418$ Å) at room temperature for the sample sealed in a silica glass capillary with an inner diameter of 0.5 mm with $\theta$ scan. PXRD patterns for **1 ⊃ O₂** and **1⊃N₂** with the synchrotron radiation ($\lambda = 0.799999(6)$ Å) were collected at SPring-8 (BL44B2)[47]. Magnetic susceptibility measurements were performed using a Quantum Design SQUID magnetometer MPMS-XL on a polycrystalline sample in the temperature range of 1.8–300 K at a dc field of 1 kOe. Diamagnetic contributions were collected for the sample holder, Nujol, and for the sample using Pascal's constants[48]. Fresh samples taken immediately from the stock liquids were used for the magnetic measurements of **1-solv**, and the formula determined by single-crystal X-ray crystallography was used for data analyses. Details for in situ IR spectra and gas adsorption-magnetic measurements are described in Supplementary Methods.

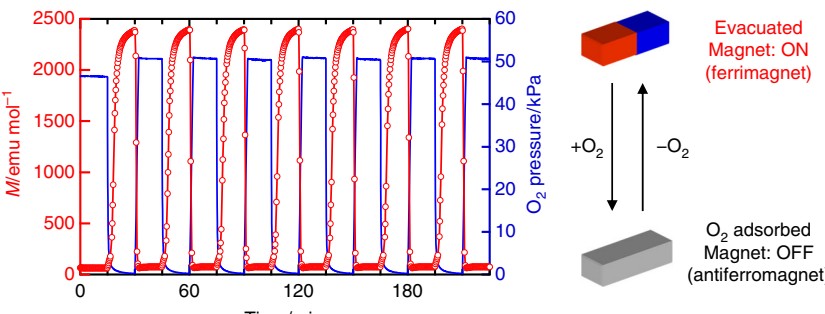

**Fig. 6** Magnetic switching by introducing/evacuating $O_2$ gas into/from **1**. Time course of the magnetization (red circle) while the $O_2$ pressure (blue line) alternated between <0.1 and 50 kPa every 15 min under a 100 Oe magnetic field at 85 K

**X-Ray crystallographic analysis for 1-solv, 1, 1 ⊃ N₂, 1 ⊃ O₂-I, and 1 ⊃ CO₂.** Crystal data for **1-solv, 1, 1 ⊃ N₂, 1 ⊃ O₂-I**, and **1 ⊃ CO₂** were collected at 134 K, 112 K, 130 K, 130 K, and 195 K, respectively, on a CCD diffractometer (Rigaku Saturn724) with multi-layer mirror monochromated Mo-Kα radiation (λ = 0.71075 Å). Details for the measurements and structural determination are described in Supplementary Methods. These data have been deposited as CIFs at the Cambridge Data Centre as supplementary publication nos. CCDC-1519242, 1519241, 1519243, 1519244, and 1519240 for **1-solv, 1, 1 ⊃ N₂, 1 ⊃ O₂-I**, and **1 ⊃ CO₂**, respectively. Structural diagrams were prepared using VESTA software[49]. The void volumes in the crystal structures were estimated using PLATON[50].

**Gas adsorption measurements.** The sorption isotherm measurements for N₂ (at 77 and 120 K), O₂ (at 90 and 120 K), and CO₂ (at 195 K) gas were performed using an automatic volumetric adsorption apparatus (BELSORP max; BEL Inc). A known weight (ca. 30 mg) of the dried sample was placed into the sample cell and then, prior to measurements, was evacuated using the degas function of the analyzer for 12 h at 353 K. The change in pressure was then monitored and the degree of adsorption was determined by the decrease in pressure at the equilibrium state.

**Gas atmosphere PXRD measurements and Structural determination of 1 ⊃ O₂-II.** A ground sample of **1** was sealed in a silica glass capillary with an inner diameter of 0.5 mm. The PXRD pattern was obtained with a 0.02° step using an Ultima IV diffractometer with Cu-Kα radiation (λ = 1.5418 Å) with θ scan. To obtain the PXRD patterns under the gas-adsorbed conditions, the glass capillary was connected to stainless-steel (SUS) lines with valves to dose and remove the gas, which were connected to a gas-handling system (BELSORP max; BEL inc). The temperature was controlled by a N₂ gas stream. Structures are determined using DIFFRACplus TOPAS® v4.2 software, FOX software[51], and RIETAN-FP software[52]. Details for structural determination are described in Supplementary Methods. These data have been deposited as CIFs at the Cambridge Data Centre as supplementary publication nos. CCDC-1519245.

## Data availability
The data sets generated during and/or analyzed during the current study are available from the corresponding author on reasonable request. The X-ray crystallographic coordinates for structures reported in this study have been deposited at the Cambridge Crystallographic Data Centre (CCDC), under deposition numbers 1519240-1519245. These data can be obtained free of charge from The Cambridge Crystallographic Data Centre via www.ccdc.cam.ac.uk/data_request/cif.

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

## Acknowledgements

We thank Dr. Hiroyasu Sato (Rigaku Co.Ltd.) for his help in structural determinations for the gas-accommodated compounds. This work was supported by a Grant-in-Aid for Scientific Research (Grant No. 16H02269, 15K13652, 26810029, and 18K05055), a Grand-in-Aid for Scientific Research on Innovative Areas ('π-System Figuration' Area 2601, Grant No. 15H00983), and a Grand-in-Aid for Specially Promoted Research (Grant No. 25000007) from the Ministry of Education, Culture, Sports, Science, and Technology, Japan (MEXT), the E-IMR project, the Asahi Glass Foundation, Mitsubishi Foundation, and the Support Program for Interdisciplinary Research in Tohoku University. J.Z. is thankful for the JSPS Research Fellowship for Young Scientists (No. 17J02497). iCeMS is supported by the World Premier International Research Institutive (WPI) of MEXT.

## Author contributions

W.K. and H.M. formulated the project. Z.L. and J.Z. synthesized and characterized the compounds. W.K. performed the gas sorption measurements and the in situ study under gases in the magnetic studies, IR spectroscopy, and PXRD measurements with the refinement of structures. J.Z. performed SCXRD measurements under gases. Y.S. and A.H. conducted in situ PXRD measurements with the synchrotron radiation under gases under the supervision of S.K. and R.M.; W.K. and H.M. wrote the manuscript and all authors discussed the results and revised the paper.

## Additional information

**Competing interests:** The authors declare no competing interests.

