## [Peer Review File · Nature Communications]

Reviewers' comments:

Reviewer #1 (Remarks to the Author):

The results from Miyasaka and co-workers are very relevant and perfectly suited for publication in Nature Communications. The paper describes a responsive porous magnet that changes its magnetic phase when adsorbing different gases, thus permitting to distinguish between paramagnetic and diamagnetic molecules, exemplified with O₂ vs N₂ or CO₂. Although some previous reports have shown that sorption of O₂ can cause a response in the magnetic properties of a MOF (JACS 2008, 130, 16921), this manuscript presents a different approach with no particular binding site in the framework, but through-space interaction. It would be nice to see further examples with other magnetic molecules such as NO.

The paper is technically very well conducted, specially with X-ray powder diffraction studies under gas pressure and also in-situ magnetic measurements with gas pressure, which are uncommon. These provide direct evidence of the magnetic ordering of the oxygen molecules.

There are only some minor corrections that should be considered before publication:

1. The Rietveld refinement shown in Figure S6c (1@O₂ at 94K) should be revised. In the region 20-60 degrees the calculated pattern (black line) describes very poorly the experimental pattern, with too broad peaks that do not correspond to the sharp peaks observed experimentally. This contrasts with the other three Rietveld refinements, which are much better performed.
2. Figure 2c shows the structural changes upon sorption of different gases. The interlayer distance could be indicated here, as is discussed in the text to be important for the accommodation of larger number of gas molecules in the case 1@CO₂ and 1@O₂-II.

Reviewer #2 (Remarks to the Author):

This resubmitted manuscript contains a complete revamp of the magnetic measurements and analyses, and this is now of much higher quality as well as being much easier to follow due to some greater attention having been given to the identity of the particular phases measured and to the matching of text with the figures. Most significantly, a phase diagram has now been mapped out that demonstrates that the magnetic changes occur independently of the structural phase change – given the subtlety of relationships between structure and magnetism there may still be some small remaining uncertainty as to the relative importance of trigger (iii), but nonetheless there is now a much higher level of insight on what's going on magnetically, with some accompanying changes to some of the interpretations. I will leave detailed assessment of the magnetic data to a more expert assessor and focus my attention on the other aspects.

In mapping out phase space, a large body of new lab (Fig S13) and synchrotron data (Fig. S14) has been collected, which is very helpful in giving insight into the gas- and temperature-dependent changes. It is something of a missed opportunity that these haven't been fitted to yield cell parameters and are presented only for qualitative comparison, but I recognize that doing this might be difficult and time-consuming.

While the new data is a welcome addition, I remain quite uncomfortable with many aspects of the powder diffraction structural work, which is largely unchanged from the previous submission. Contrary to the rebuttal's rather dismissive and quite unaccommodating response (in which I'm misquoted in places – eg. I did not refer to the data as "dirty"), I am not advocating that they give up on this; rather, I think these are important but that it is essential that an appropriately balanced and informative analysis and description is given, which I feel is still not the case.

Specifically, I believe the current presentation of these structures leads the reader to assign a greater level of certainty than is warranted. The following recommendations address this.

Most fundamentally, it remains that the reader has no real way to assess the veracity of the Rietveld refined structures. Indeed, it seems a misnomer to label these as “Rietveld refined” when they appear to be fully fixed structural models; only 7 parameters are refined, which are presumably the unit cell and peak width parameters(?) – basically these ‘refined’ structures in reality might more appropriately be described as ‘simulated’. The SI should state exactly how these refinements were performed – what parameters were in fact fitted? The experimental states “Soft constraints on bond distances, bond angles, and dihedral angles were adopted throughout the refinement” – if so why submit unrefined models? In short, it wouldn’t surprise me if quite different structural arrangements gave similarly good fits to the data.

The rebuttal states “we also know that the data for high symmetrical framework structures such as our cases relatively give good analytic quality in information on the basis of framework structure and its packing form in a crystal”. This is not a high symmetry phase. It is in the lowest crystal system, triclinic, in which very particular care is needed to avoid peak misassignment and false minima in structure elucidation/refinement. There are two possible space groups for the gas loaded structures, P1 and P-1, with it being assumed that the latter is correct; these are the lowest symmetry space groups.

The rebuttal states “what we have to refine to obtain initial structural model is only the orientation of each molecular unit (two [Ru2] unit and TCNQ) constrained around each inversion center”. Beyond the point that there is no evidence that the space group is P-1 and not P1 for the XRPD phases, this is a fundamental misconception in the role of symmetry operations. They are not invisible physical constraints that exist within crystals. Crystallographic refinement software may treat them this way once it is established that a molecule sits on a symmetry site, but this is not the same as being allowed to state that “each unit of [Ru2] units and TCNQ has an inversion center”. A sizeable proportion of molecular structures contain symmetry-generated disorder, and believing that the gas-loaded phases aren’t among them is an assumption that is currently unstated.

The XRPD fits have been made against poor quality lab data, and now that higher quality synchrotron data exist there is the opportunity to determine just how good these models are – the higher quality data will be much less forgiving of inaccuracies in these models, eg. as would be the case if peak merging of the broadened lab data allowed misassignment of intensities, and quite possibly also peak indexing. I won’t go as far as to advocate that all of the structures are re-refined against the synchrotron data (this is a lot of work), but the fact that the authors have apparently done this for the first O2 phase but then chosen not to provide any details doesn’t inspire confidence, nor does the fact that the relative peak intensities are quite different between the lab and synchrotron data (for example, see eg. Fig S14a; here the intensity of the peaks from 5-5.5 degrees actually show the reverse trend of the lab data models). This suggests to me that the lab and synchrotron data will lead to quite different structural models. Just how good was the agreement? This needs to be shown to demonstrate consistency across different measurements.

One measure of the veracity of the models is whether the atom-atom contacts are reasonable, and as I have pointed out previously this is not the case, and still isn’t (the structures are unchanged) – numerous unphysical interactions exist between the lattice and guests (see numerous high level alerts in the cifchecks), and it is additionally quite unexpected that the different guests occupy quite different regions of the pores. This is a point of central importance given the proposed role of the O2 molecules in influencing the magnetic properties; in both O2 structures the O2 molecules lie in unphysical positions with unacceptably close contact with the framework. This is really not a good thing to get wrong. Given that the authors are persisting in showing the guest-loaded structures (now moved to the SI) and state that “The introduction of gas molecules into the frameworks is absolutely imperative to construct a plausible initial structural model that satisfies

the observed PXRD pattern by a real-space method", I would suggest that it is important that they display the electron difference density maps for these structures with the guests removed so that the reader has the ability to gauge whether these pictures should be regarded as accurate. In particular, they should compare the difference density plots for the lab and synchrotron O₂ datasets to demonstrate consistency.

In responding to the rebuttal statement "We suppose what most of readers require from the structural data obtained by this technique is as follows: an overview of the framework packing with inter-layer distances, the relative configuration of each unit (connection between donor and acceptor), and cell information (volume change). In this sense, we judged that large esds values for coordinates are useless information", I would just reemphasise that readers foremost require the capability to gauge the accuracy of the structures, for which esds are one of many measures; without this, the rest is meaningless.

On the issue of the conflation of separation and recognition, I respectfully disagree, and would point out that the conflation of these distinct processes has led to reviewer 3 – a confessed non-expert in magnetism - mistakenly thinking that the significance of the work is in O₂/N₂ separation. I feel it is essential that this point is clear, particularly given the general readership of this journal, and am mystified as to why the authors are unwilling to reframe their work with some rewording. While the change in title has led to an improvement, the introduction text still reads: "For instance, distinguishing between nitrogen (N₂) and oxygen (O₂) gases is exceedingly difficult because of their similar size and boiling points.³ Currently, large-scale apparatuses such as cryogenic gas separations or adsorption separations that exploit their small boiling point or kinetic diameter differenced are used for separating N₂ and O₂;⁴ however, these techniques consume large amounts of energy. Detecting a magnetic change induced by the intrinsic magnetic nature of these gases (i.e., diamagnetic N₂ and paramagnetic O₂) would represent a major breakthrough in gas-sensing technologies." To the non-expert this is very misleading.

Lastly, I'd like to request that much greater attention be given to the figure and table captions, making it clear what instrument (eg. lab or synchrotron xrd; powder or single crystal; etc.) and conditions (eg. gas pressure) were used. Absence of this makes the work very difficult to follow.

Answers and comments to the reviewer's comments

We appreciate the editors and reviewers very much for their constructive comments and suggestions on this manuscript (NCOMMS-18-18061-T). We have carefully and thoroughly revised our manuscript considering relevant concerns given by reviewers. The details of our specific comments are as follows, where the answer/comments from us are given in blue:

Reviewer #1

The results from Miyasaka and co-workers are very relevant and perfectly suited for publication in Nature Communications. The paper describes a responsive porous magnet that changes its magnetic phase when adsorbing different gases, thus permitting to distinguish between paramagnetic and diamagnetic molecules, exemplified with O₂ vs N₂ or CO₂. Although some previous reports have shown that sorption of O₂ can cause a response in the magnetic properties of a MOF (JACS 2008, 130, 16921), this manuscript presents a different approach with no particular binding site in the framework, but through-space interaction. It would be nice to see further examples with other magnetic molecules such as NO.

The paper is technically very well conducted, specially with X-ray powder diffraction studies under gas pressure and also in-situ magnetic measurements with gas pressure, which are uncommon. These provide direct evidence of the magnetic ordering of the oxygen molecules.

>> Thank you for a favorable comments.

There are only some minor corrections that should be considered before publication:
1. The Rietveld refinement shown in Figure S6c (1@O₂ at 94K) should be revised. In the region 20-60 degrees the calculated pattern (black line) describes very poorly the experimental pattern, with too broad peaks that do not correspond to the sharp peaks observed experimentally. This contrasts with the other three Rietveld refinements, which are much better performed.

>> Firstly, we have to emphasize that the structures of 1@CO₂, 1@N₂, and 1@O₂-I were finally determined by single-crystal X-ray diffraction (SCXRD) analyses under pressure-controlled gas atmospheres, which drastically improved the reliability of their structures that was crucial for the principal discussion in this work, i.e., a comparison between 1@N₂, and 1@O₂-I.

Unfortunately, SCXRD measurement for 1@O₂-II was failed because of its poor single-crystallinity under higher O₂ pressures. Hence, only the indexing result for 1@O₂-II is

nonempirical (only PXRD pattern with the Rietveld refinement for **1@O₂-II** was remained in Figure S8). Nevertheless, we concluded that the obtained structure is reliable at least for the discussion on its cell-constants, volume, and interlayer distance, which was confirmed by comparing obtained structural parameters with those of the other gas-adsorbed phases; see the following figure on PXRD indexing results.

Since our compounds have a similar layered form even under gas atmospheres, where the [Ru₂] dimer and μ₄-TCNQ are alternately located, we can recognize an order in the Miller indices of the diffraction peaks with strong intensity, despite a triclinic system. This “pseudo-extinction rule” helps a lot for indexing of PXRD patterns in our systems. In light of the rule, we determined the lattice constants of **1@O₂-II**. Importantly, we could obtain a well-reasonable [Ru₂]₂-TCNQ layered structure from the PXRD pattern for **1@O₂-II** and the above indexes, though the fitting quality was worse compared to others. The inaccurate information on the position and number of O₂ molecules in **1@O₂-II** made its PXRD simulation much poorly, so we cannot discuss on the accommodated O₂ molecules in **1@O₂-II**.

2. Figure 2c shows the structural changes upon sorption of different gases. The interlayer distance could be indicated here, as is discussed in the text to be important for the accommodation of larger number of gas molecules in the case 1@CO₂ and 1@O₂-II.

>> According to the referee's comments, one of the two types of interlayer distances, vertical distance (l_1) was added in Figure 2c. Another interlayer distance, i.e., the inter-unit translational distance (l_2) corresponds to the a or a' unit cell axis, which was commented in the figure caption. These inter-layer distances were also added in Figure 1b and 1d for **1-solv** and **1**.

Reviewer #2

This resubmitted manuscript contains a complete revamp of the magnetic measurements and analyses, and this is now of much higher quality as well as being much easier to follow due to some greater attention having been given to the identity of the particular phases measured and to the matching of text with the figures. Most significantly, a phase diagram has now been mapped out that demonstrates that the magnetic changes occur independently of the structural phase change – given the subtlety of relationships between structure and magnetism there may still be some small remaining uncertainty as to the relative importance of trigger (iii), but nonetheless there is now a much higher level of insight on what's going on magnetically, with some accompanying changes to some of the interpretations. I will leave detailed assessment of the magnetic data to a more expert assessor and focus my attention on the other aspects.

>> Thank you for favorable comments on the principle subject of this work.

In mapping out phase space, a large body of new lab (Fig S13) and synchrotron data (Fig. S14) has been collected, which is very helpful in giving insight into the gas- and temperature-dependent changes. It is something of a missed opportunity that these haven't been fitted to yield cell parameters and are presented only for qualitative comparison, but I recognize that doing this might be difficult and time-consuming.

>> Thank you for comprehensive comments. The main purpose of taking many data of PXRD given in Figure S13 and S14 (old) is to estimate the O_2 pressure at each temperature changing from **1 \rightarrow O₂-I** to **1 \rightarrow O₂-II**. So it was strictly, accurately determined from them, although it was done without any fitting of PXRD patterns.

While the new data is a welcome addition, I remain quite uncomfortable with many aspects of the powder diffraction structural work, which is largely unchanged from the previous submission. Contrary to the rebuttal's rather dismissive and quite unaccommodating response (in which I'm misquoted in places – eg. I did not refer to the data as “dirty”), I am not advocating that they give up on this; rather, I think these are important but that it is essential that an appropriately balanced and informative analysis and description is given, which I feel is still not the case. Specifically, I believe the current presentation of these

structures leads the reader to assign a greater level of certainty than is warranted. The following recommendations address this.

Most fundamentally, it remains that the reader has no real way to assess the veracity of the Rietveld refined structures.

>> We finally found the most convincing answer to this comment; we succeeded in revealing the structures for **1**CO₂, **1**N₂, and **1**O₂-I taken by single-crystal X-ray diffraction (SCXRD) analyses under pressure-controlled gas atmospheres; the experimental details were noted in SI and the structural information for these phases are added/modified in Figures 2c, S6, S7, S9, and S10, Table S4– S6, and S8. It is, in general, difficult to apply single-crystal technique for gas-adsorbed phase because of a degradation of single crystal quality. However, we finally found an efficient SCXRD technique applicable for our system. These structural data, especially those of **1**N₂, and **1**O₂-I, are critical for discussing the principal subject on the magnetic difference between isostructural series of **1**N₂, and **1**O₂-I, although the structure of **1**O₂-II was not isolated (vide infra). We have to note here that the PXRD patterns with the Rietveld atomic assignments for the frameworks were in completely agreement with the PXRD simulated patterns estimated from SCXRD, and hence, the overall story of the manuscript was remained. The PXRD patterns for **1**CO₂, **1**N₂, and **1**O₂-I are no longer necessary in this work, so they were removed from SI.

The relevant descriptions and data newly given are displayed below:

Experimental details in SI

*A single crystal of **1-solv** was mounted on a thin glass capillary with a minimum amount of epoxy adhesive, which was attached to the inner-wall of silica glass capillary with an inner diameter of 0.5 mm. The capillary was connected to gas-handling system through o-ring seal connector based on Swagelok Ultra-Torr[®] adaptor, which can be fixed on x-ray goniometer. Then the crystal was evacuated at room temperature for one hour, 100 kPa of N₂, 10 kPa of O₂, and 100 kPa of CO₂ was introduced at room temperature, then cooled by N₂ gas stream.*

Figure S7. Thermal ellipsoid plot obtained from single crystal XRD analyses showing the asymmetric unit and atom numbering scheme for **1**⊃**N**₂ (a), **1**⊃**O**₂-**I** (b), and **1**⊃**CO**₂ (c). Displacement ellipsoids are drawn at a 50% probability level. Gray bond in (b) indicates the minor component of the rotational disorder. Hydrogen atoms and guest molecules are omitted for clarity. Symmetry codes: (*) $-x, -y, -z$, (**) $-x + 1, -y + 1, -z + 1$, (***) $-x, -y, -z + 1$.

Table S4. Crystallographic data obtained from single crystal XRD analyses for **1D_{N2}**, **1D_{O2-I}**, and **1D_{CO2}**.

	1D_{N2}	1D_{O2-I}	1D_{CO2}
formula	C ₇₀ H ₃₂ F ₁₆ N ₁₂ O ₁₈ Ru ₄	C ₇₀ H ₃₂ F ₁₆ N ₄ O _{28.3} Ru ₄	C ₇₅ H ₃₂ F ₁₆ N ₄ O ₂₈ Ru ₄
formula weight	2037.35	2090.09	2145.34
crystal system	Triclinic	Triclinic	Triclinic
space group	P -1	P -1	P -1
a / Å	10.5642(18)	10.6878(15)	10.8173(11)
b / Å	11.986(2)	12.1373(17)	12.4400(12)
c / Å	15.718(3)	15.895(2)	15.9803(15)
α / deg	91.883(15)	91.728(12)	90.830(8)
β / deg	92.336(14)	93.681(12)	96.483(8)
γ / deg	103.024(16)	103.496(12)	104.721(9)
V / Å ³	1935.6(7)	1998.7(5)	2064.4(4)
Z	1	1	1
crystal size / mm ³	0.13×0.05×0.03	0.11×0.06×0.04	0.14×0.04×0.02
T / K	130(1)	130(1)	195(1)
D _{calc} / g·cm ⁻³	1.748	1.736	1.725
F ₀₀₀	1000.00	1026.40	1054.00
λ / Å	0.71073	0.71073	0.71073
μ (Mo K α) / cm ⁻¹	8.799	8.610	8.359
data measured	12560	13334	13705
data unique	6894	7157	7394
R _{int}	0.0555	0.0512	0.0399
no. of observations	6894	7157	7394
no. of variables	541	632	615
R 1 (I > 2.00 σ (I)) ^a	0.1194	0.1080	0.0808
R (all reflections) ^a	0.1776	0.1814	0.1334
wR 2 (all reflections) ^b	0.3067	0.2893	0.2308
GOF	1.137	1.057	1.119
CCDC No.	1519243	1519244	1519240

^a $R1 = R = \sum ||F_o| - |F_c|| / \sum |F_o|$. ^b $wR2 = [\sum w(F_o^2 - F_c^2)^2 / \sum w(F_o^2)^2]^{1/2}$

A comparison of experimental PXRD patterns and the simulated ones evaluated from single crystal structures were added as Figure S7.

Figure S7. Comparison of observed PXRD patterns and calculated PXRD patterns from the structures obtained from single crystal XRD analyses.

It should be commented here that the position of accommodated gases was not the same between the single crystal structures and the Rietveld ones; gases have actually been mislocated in the Rietveld structures. However, the accommodated gases were quite reasonably located in this improvement, determined by SCXRD analyses for **1**⊃CO₂, **1**⊃N₂, and **1**⊃O₂-I, which were shown in Figure 2c and Figure S9 (vide infra).

The accommodated gases for **1**⊃O₂-II, which were introduced for the construction of initial model for the Rietveld refinement, were not shown and discussed in the manuscript.

Figure 2. Gas sorption capability, sorption-induced structural transitions, and crystal

structures under gases. **a**, Adsorption (closed) and desorption (open) isotherms of **1** for CO₂ at 195 K (green), N₂ at 77 K (orange) and 120 K (red), O₂ at 90 K (blue) and 120 K (cyan), where solid lines are only guide for the eye. **b**, PXRD patterns measured ($\lambda = 1.54 \text{ \AA}$) under 100 kPa of N₂ at 130 K (red), O₂ at 130 K (cyan) and 94 K (blue) and CO₂ at 204 K (green) and ones for **1** (black) and **1-solv** (purple) at 298 K. **c**, Packing views (c' -axis projection, where atoms N, O, C, F, and Ru are represented in blue red, gray, green, and purple, respectively) of **1**⊃N₂ at 130 K, (upper left), **1**⊃O₂-**I** at 130 K (lower left), **1**⊃O₂-**II** at 94 K (lower right), and **1**⊃CO₂ at 195 K (upper right) and their transformation routes including **1-solv** and **1**, where the structures for **1**⊃N₂, **1**⊃O₂-**I**, and **1**⊃O₂-**II** were determined by *in situ* SCXRD analyses, and it for **1**⊃O₂-**II** was evaluated from the PXRD pattern with the Rietveld refinement (the accommodated gas molecules in a space filling model, of which the site occupancy is colored, are only represented in **1**⊃N₂, **1**⊃O₂-**I**, and **1**⊃O₂-**II**, and minor position-disordered components in a part of layer framework in **1**⊃O₂-**I** were omitted for clarity). The a' - c' axes represent those of a transformed lattice for easy comparison with the lattice of **1-solv** (see Supplementary). The vertical inter-layer distances (l_1) are indicated in blue digit, and the inter-unit translational distance (l_2) corresponds to the a' -lattice axis for each compound (Table S8). **d**, A close-up view of gas-sandwiched mode at Site A (Fig. S10) denoted as an yellow ellipse in figure c for **1**⊃O₂-**I** and **1**⊃N₂.

Figure S9. Packing views of $1\text{D}\text{N}_2$ (a), $1\text{D}\text{O}_2\text{-I}$ (b), and $1\text{D}\text{CO}_2$ (c) obtained from single-crystal XRD analyses along the a' -axis, where atoms N, O, C, F and Ru are represented in blue red, gray, green and purple, respectively, and the adsorbed gas molecules that were located by the structural analyses are given in space filling model, where the colored area fraction of the atomic sphere corresponds to site occupancy. The axes in red represent a transformed lattice for an easy comparison with the lattice of 1-solv . Minor component of positional disordered atoms of the benzoate moiety in $1\text{D}\text{O}_2\text{-I}$ were omitted for clarity.

Gases were, commonly in $1\text{D}\text{CO}_2$, $1\text{D}\text{N}_2$, and $1\text{D}\text{O}_2\text{-I}$, accommodated at distinguishable three sites (see below, Figure S10); A-site is a position sandwiched by π -planes of the TCNQ(OMe)₂ moieties, B-site is adjacent to the methoxy group of TCNQ(OMe)₂, and C-site is located between B-sites, where B- and C sites are in hexagonal pores of fishnet frameworks.

Judging from the site occupancy in SCXRD analyses, N_2 and O_2 gases preferred A-site, while disordered CO_2 molecules with a small site occupancy were found at A-site.

Conversely, CO₂ gases could prefer B- and C-sites probably. N₂ gases were also found at B-site, but not at C-site. Disordered O₂ gases were also recognized among B- and C-sites.

Relevant structural dimensions around A-site for **1**⊃O₂-I and **1**⊃N₂ were shown in Figure 2d. Judging from the location of gases, O₂ gases at A-site are expected to be associated with the magnetic pathway discussed in this work. Compound **1**⊃N₂ also had almost the same molecular arrangement at A-site as **1**⊃O₂-I. This structural aspect on the position of gases strongly supports the discussion on antiferromagnetic ordering via O₂ spins, which was only found in **1**⊃O₂-I. These descriptions were added in the revised manuscript and SI.

Figure S10. Schematic diagram of gas adsorption site for **1**⊃N₂, **1**⊃O₂-I, and **1**⊃CO₂ along a' -axis (left) and c' -axis (right).

The comparisons between the structures determined by SCXRD and PXRD with Rietveld refinement are shown below (as review-only material).

Table. Relevant crystallographic data for **1**⊃N₂.

	1 ⊃N ₂ (Single crystal)	1 ⊃N ₂ (Rietveld)
a / Å	10.5642(18)	10.6149(8)
b / Å	11.986(2)	12.0488(10)
c / Å	15.718(3)	15.7483(8)
α / deg	91.883(15)	91.624(7)
β / deg	92.336(14)	91.650(8)
γ / deg	103.024(16)	103.451(6)
V / Å ³	1935.6(7)	1956.8(3)
T / K	130(1)	130(1)
Guest accessible volume / Å ³	335.3 (17.3%)	343.3 (17.5%)
Inter-layer vertical distance l ₁ / Å	8.76	8.83

**Figure.** Structural comparison for **1**⊃N₂.

Table. Relevant crystallographic data for **1 \supset O₂-I**.

	1\supsetO₂-I (Single crystal)	1\supsetO₂-I (Rietveld)
$a / \text{\AA}$	10.6878(15)	10.6472(12)
$b / \text{\AA}$	12.1373(17)	12.0642(14)
$c / \text{\AA}$	15.895(2)	15.6951(13)
α / deg	91.728(12)	91.684(10)
β / deg	93.681(12)	91.300(12)
γ / deg	103.496(12)	103.573(9)
$V / \text{\AA}^3$	1998.7(5)	1958.0(4)
T / K	130(1)	130(1)
Guest accessible volume / \AA^3	369.3 (18.5%)	289.9 (14.8%)
Inter-layer vertical distance $l_1 / \text{\AA}$	8.89	8.86

**Figure.** Structural comparison for **1 \supset O₂-I**.

Table. Relevant crystallographic data for **1**⊃**CO**₂.

	1 ⊃ CO ₂ (Single crystal)	1 ⊃ CO ₂ (Rietveld)
a / Å	10.8173(11)	10.7715(8)
b / Å	12.4400(12)	12.4658(9)
c / Å	15.9803(15)	15.9248(9)
α / deg	90.830(8)	90.863(6)
β / deg	96.483(8)	96.553(7)
γ / deg	104.721(9)	104.883(6)
V / Å ³	2064.4(4)	2050.0(2)
T / K	195(1)	204(1)
Guest accessible volume / Å ³	447.9 (21.7%)	395.8 (19.3%)
Inter-layer vertical distance l ₁ / Å	9.10	9.10

**Figure.** Structural comparison for **1**⊃**CO**₂.

Concerning **1 \supset O₂-II**, SCXRD analysis was unfortunately failed because the crystallinity of crystal sample was lost when a high pressure of O₂ gas was loaded at low temperatures. However, by comparing the obtained structural parameters with those of the other gas-adsorbed phases, we got much higher reliability on the framework structure of **1 \supset O₂-II** evaluated by the Rietveld refinement for PXRD data. At least, it is reliable for the discussion of cell-constants, volume, and interlayer distance on **1 \supset O₂-II**.

Our compounds have an identical layered structure, in which [Ru₂] dimer and μ_4 -TCNQ are alternately located, even in compounds gas accommodated, so we can recognize an order in the Miller indices of the diffraction peaks with strong intensity, despite triclinic system. This “pseudo-extinction rule” helps a lot for indexing PXRD patterns in our systems. In light of the above rule, we have determined the lattice constants of **1 \supset O₂-II**.

Indeed, it seems a misnomer to label these as “Rietveld refined” when they appear to be fully fixed structural models; only 7 parameters are refined, which are presumably the unit cell and peak width parameters(?) – basically these ‘refined’ structures in reality might more appropriately be described as ‘simulated’. The SI should state exactly how these refinements were performed – what parameters were in fact fitted? The experimental states “Soft constraints on bond distances, bond angles, and dihedral angles were adopted throughout the refinement” – if so why submit unrefined models?

>> Refined parameters were as follows; a) Three peak-shift parameters, b) twelve background parameters, c) scale factor, d) peak profile parameters with Split-Pearson VII

function (three peak width parameters, three asymmetry parameters, and four decay parameters), e) lattice constants (a , b , c , α , β , γ), f) overall isotropic atomic displacement parameter, and g) fractional coordinates (x , y , z) for non-hydrogen atoms (~60 crystallographically unique non-hydrogen atoms). All of these parameters were refined under soft constraints to maintain chemically-meaningful structures. On the final LSQ cycles, the above parameters except for six lattice constants and scale factors were fixed to evaluate standard error of the lattice constants. Hence, the given structures were not “simulated”, but “refined”.

The parameters were refined incrementally with a conjugate-direction method. Refined parameters in each step are as follows (Scale factor was refined in the all step):

1. Peak-shift and background parameters
2. Lattice constants
3. Peak-shift, background parameters, lattice constants
4. Peak width parameters
5. Peak asymmetry parameters
6. Peak decay parameters
7. Peak width parameters
8. Peak width and asymmetry parameters
9. Peak width, asymmetry, and decay parameters
10. Peak-shift, background parameters, and lattice constants
11. Fractional Coordinates for C1-7, F1-2, O1-O2 (Benzoate moiety 1)
12. Fractional Coordinates for C8-14, F3-4, O3-O4 (Benzoate moiety 2)
13. Fractional Coordinates for C15-21, F5-6, O5-O6 (Benzoate moiety 3)
14. Fractional Coordinates for C22-28, F7-8, O7-O8 (Benzoate moiety 4)
15. Fractional Coordinates for N1-2, C29-35, O9 (TCNQ(OMe)₂)
16. Fractional coordinates for Ru1-2 and gases
17. Fractional coordinates for all atoms
18. Peak-shift, background parameters, and lattice constants
19. Peak-shift, background parameters, lattice constants, and Fractional coordinates
20. Peak-shift, background, peak profile parameters, and lattice constants
21. Peak-shift, background, peak profile parameters, lattice constants and Fractional coordinates
22. Isotropic atomic displacement parameter
23. Peak-shift, background, peak profile parameters, and lattice constants
24. Peak-shift, background, peak profile parameters, lattice constants and Fractional coordinates

In the initial stage of the refinement, hydrogen atoms were removed from the structural model. After all parameters were refined, hydrogen atoms were attached to the calculated positions, and then, all parameters were refined. Fractional coordinates of hydrogen atoms were not refined. After the refinement, fractional coordinates of hydrogen atoms were again calculated and modified. This process was repeated until the fractional coordinates of hydrogen atoms became self-consistent. On the final refinement, all parameters except for lattice constants and scale factor were fixed to evaluate standard error of the lattice constants. The description about the fitted parameters and the essence of the refinement procedures are added to the SI.

In short, it wouldn't surprise me if quite different structural arrangements gave similarly good fits to the data.

>> This is just preconception. On the stage of the initial model construction, we used a direct space method with a Monte-Carlo simulation, where the orientation of [Ru₂] and TCNQ models are varied. Among the obtained results after hundreds of trials, 20~50% of the structures were satisfactory from the viewpoint of the bonding form of [Ru₂] unit and TCNQ, i.e., the formation of the layered network. The possibility of mis-connecting (i.e., mis-bonding) form of layered structure should be discarded from the data of other physical properties. Importantly, the satisfactory candidates of structures mentioned above were almost identical to each other. Hence, there is no chance of "quite different structural arrangements gave similarly good fits" in our system. Actually, if we met a situation on "quite different structural arrangements gave similarly good fits", we would not have planned to determine crystal structures from PXRD.

The rebuttal states "we also know that the data for high symmetrical framework structures such as our cases relatively give good analytic quality in information on the basis of framework structure and its packing form in a crystal". This is not a high symmetry phase. It is in the lowest crystal system, triclinic, in which very particular care is needed to avoid peak misassignment and false minima in structure elucidation/refinement. There are two possible space groups for the gas loaded structures, P1 and P-1, with it being assumed that the latter is correct; these are the lowest symmetry space groups.

>> The triclinic space group is, in common, not a high symmetric phase, but the fishnet framework in this type of compounds has a high symmetric form, most of which revealed an identical structure crystallized in the triclinic *P*-1 space group with an inversion center (*Z* = 1). Actually, the SCXRD analyses for **1**⊃CO₂, **1**⊃N₂, and **1**⊃O₂-I revealed the *P*-1 space

group, which strongly supported the same space group even in **1O₂-II** (there is no example of *P*1 in this type of compounds so far).

The rebuttal states “what we have to refine to obtain initial structural model is only the orientation of each molecular unit (two [Ru₂] unit and TCNQ) constrained around each inversion center”. Beyond the point that there is no evidence that the space group is *P*-1 and not *P*1 for the XRPD phases, this is a fundamental misconception in the role of symmetry operations. They are not invisible physical constraints that exist within crystals. Crystallographic refinement software may treat them this way once it is established that a molecule sits on a symmetry site, but this is not the same as being allowed to state that “each unit of [Ru₂] units and TCNQ has an inversion center”. A sizeable proportion of molecular structures contain symmetry-generated disorder, and believing that the gas-loaded phases aren’t among them is an assumption that is currently unstated.

>> In general situation, this referee’s comment could be correct, but on the basis of our experiences about this type of [Ru₂]-TCNQ compounds, we convinced that **1O₂-II** is also in the *P*-1 space group with an inversion center on each [Ru₂] unit and TCNQ. By the way, there have never been found in *P*1 in the series.

The XRPD fits have been made against poor quality lab data, and now that higher quality synchrotron data exist there is the opportunity to determine just how good these models are – the higher quality data will be much less forgiving of inaccuracies in these models, eg. as would be the case if peak merging of the broadened lab data allowed misassignment of intensities, and quite possibly also peak indexing.

>> Now the single crystal structures were obtained for **1CO₂**, **1N₂**, and **1O₂-I**. The structure even for **1O₂-II** could be evaluated based on the SCXRD data of **1CO₂**, **1N₂**, and **1O₂-I** to strongly convince its accuracy on the structure. So let us skip to reply this question.

I won’t go as far as to advocate that all of the structures are re-refined against the synchrotron data (this is a lot of work), but the fact that the authors have apparently done this for the first O₂ phase but then chosen not to provide any details doesn’t inspire confidence, nor does the fact that the relative peak intensities are quite different between the lab and synchrotron data (for example, see eg. Fig S14a; here the intensity of the peaks from 5-5.5 degrees actually show the reverse trend of the lab data models). This suggests to me that the lab and synchrotron data will lead to quite different structural models. Just

how good was the agreement? This needs to be shown to demonstrate consistency across different measurements.

>> Now we obtained structures for 1CO_2 , 1N_2 , and $1\text{O}_2\text{-I}$ based on SCXRD data. Here we have to emphasize that PXRD patterns for 1CO_2 , 1N_2 , and $1\text{O}_2\text{-I}$ were identical to their simulated patterns and the model structures evaluated from PXRD patterns were basically the same as the SCXRD structures except for the position and number of some gas molecules. This conclusion completely agreed with the description/discussion in the previous manuscript. Nevertheless, all descriptions on 1CO_2 , 1N_2 , and $1\text{O}_2\text{-I}$ in the revised manuscript were replaced by the data obtained from the SCXRD analyses. Just how good was the agreement for $1\text{O}_2\text{-II}$?— this might be the next question from the reviewer. However, now we have convinced structures for 1CO_2 , 1N_2 , and $1\text{O}_2\text{-I}$ and compare the model structure of $1\text{O}_2\text{-II}$ with the SCXRD structures of 1CO_2 , 1N_2 , and $1\text{O}_2\text{-I}$, which gave us a higher reliability on the structure of $1\text{O}_2\text{-II}$ than previous at least on the layer framework structure as well as discussions on cell-constants, volume, and interlayer distance of $1\text{O}_2\text{-II}$.

We have to point out here that the most important aspect on structures under gas atmospheres is the comparison of structures between 1N_2 , and $1\text{O}_2\text{-I}$, which requires for proving the difference of magnetism; this is a critical reason why we did the further experiments on just $1\text{O}_2\text{-I}$ in the previous version.

One measure of the veracity of the models is whether the atom-atom contacts are reasonable, and as I have pointed out previously this is not the case, and still isn't (the structures are unchanged) – numerous unphysical interactions exist between the lattice and guests (see numerous high level alerts in the cifchecks), and it is additionally quite unexpected that the different guests occupy quite different regions of the pores. This is a point of central importance given the proposed role of the O₂ molecules in influencing the magnetic properties; in both O₂ structures the O₂ molecules lie in unphysical positions with unacceptably close contact with the framework. This is really not a good thing to get wrong. Given that the authors are persisting in showing the guest-loaded structures (now moved to the SI) and state that “The introduction of gas molecules into the frameworks is absolutely imperative to construct a plausible initial structural model that satisfies the observed PXRD pattern by a real-space method”, I would suggest that it is important that they display the electron difference density maps for these structures with the guests removed so that the reader has the ability to gauge whether these pictures should be regarded as accurate. In particular, they should compare the difference density plots for the lab and synchrotron O₂ datasets to demonstrate consistency.

>> Here we displayed the most important structures, which are for 1D-N_2 , and $1\text{D-O}_2\text{-I}$, with high accuracy. Everybody would understand why we had a discussion on O_2 spin mediated magnetic ordering in this work. So, skip to reply to this question.

In responding to the rebuttal statement “We suppose what most of readers require from the structural data obtained by this technique is as follows: an overview of the framework packing with inter-layer distances, the relative configuration of each unit (connection between donor and acceptor), and cell information (volume change). In this sense, we judged that large esds values for coordinates are useless information”, I would just reemphasise that readers foremost require the capability to gauge the accuracy of the structures, for which esds are one of many measures; without this, the rest is meaningless.

>> It is now needless to reply to this question because the SCXRD structures were displayed for 1D-CO_2 , 1D-N_2 , and $1\text{D-O}_2\text{-I}$ in the revised version.

Extremely saying, fractional coordinates are essentially needless for the principal discussion of this work, although they are fundamental in crystallography. If we noticed a fact of the increase of inter-layer distances induced by a gas accommodation, we could speculate that an interlayer magnetic dipole-dipole interaction should be weakened. If the cell volume change is smaller than the expected volume calculated from adsorption data, we could assume that a chemical pressure effect on the framework could be enhanced by gas adsorption. For this purpose, namely, “a comparison between before/after,” Le-Beil analyses should be enough if an assumption that its layered structure, i.e., connection between donor and acceptor, must be maintained under a gas accommodation is justified. The role of Rietveld refinement is just to prove the existence of affordable network structure against the observed PXRD patterns. In this context, we reemphasize that large esds values for coordinates are useless for the comparison, because the differences of dimensions are much significant for discussing the difference between before/after.

On the issue of the conflation of separation and recognition, I respectfully disagree, and would point out that the conflation of these distinct processes has led to reviewer 3 – a confessed non-expert in magnetism - mistakenly thinking that the significance of the work is in O_2/N_2 separation. I feel it is essential that this point is clear, particularly given the general readership of this journal, and am mystified as to why the authors are unwilling to reframe their work with some rewording. While the change in title has led to an improvement, the introduction text still reads: “For instance, distinguishing between nitrogen (N_2) and oxygen (O_2) gases is exceedingly difficult because of their similar size and boiling points.”³

Currently, large-scale apparatuses such as cryogenic gas separations or adsorption separations that exploit their small boiling point or kinetic diameter differences are used for separating N₂ and O₂; however, these techniques consume large amounts of energy. Detecting a magnetic change induced by the intrinsic magnetic nature of these gases (i.e., diamagnetic N₂ and paramagnetic O₂) would represent a major breakthrough in gas-sensing technologies.” To the non-expert this is very misleading.

>> We understand what the reviewer wants to say to this description. We removed the following sentence, which may confuse readers the principal subject of this work, namely on a “gas sensing”:

“Currently, large-scale apparatuses such as cryogenic gas separations or adsorption separations that exploit their small boiling point or kinetic diameter differences are used for separating N₂ and O₂; however, these techniques consume large amounts of energy.”

Lastly, I'd like to request that much greater attention be given to the figure and table captions, making it clear what instrument (eg. lab or synchrotron xrd; powder or single crystal; etc.) and conditions (eg. gas pressure) were used. Absence of this makes the work very difficult to follow.

>> According to the referee's comments, manuscript were thoroughly revised.

Additional Corrections:

Following these corrections mainly suggested by the reviewers, some additional corrections were also made in the revised version. For example, cif file replacements, corrections on structural parameters, Tables, references, and so on, which were marked in yellow in the revised manuscript, as well as indications in this letter.

REVIEWERS' COMMENTS:

Reviewer #1 (Remarks to the Author):

The revised manuscript has largely improved the previous version, and the inclusion of the single-crystal data of the structures under different gas atmospheres shows the structural transformations without any doubt, which previously were not that unequivocal with only X-ray powder data. The manuscript is suitable for publication. I have however one comment for the authors to check: Figure S7, which shows the X-ray powder data with different gases (experimental vs calculated from single crystal diffraction), seems to have the wrong experimental powder diffractogram for O₂ at 130 K. The blue diffractogram (obs.) seems too similar to the red diffractogram (obs), which corresponds to N₂ at 130K. This is more evident by comparing with the calculated diffractograms, focussing in the region 14-16 degrees.

Reviewer #2 (Remarks to the Author):

The revised manuscript has successfully addressed all of the I raised in my previous review and in my view the manuscript is now suitable for publication – indeed, I think it will be an excellent addition to the literature. Most importantly, the authors provide single crystal X-ray structural determinations of 3 of the 4 gas-adsorbed phases, which has provided an enormous improvement in the structural understandings; notably, the concerns I had raised previously regarding the unphysical location of the gas molecules within the pores appear to have been justified, with the new single crystal determinations providing an entirely different picture of the host-guest configurations than those that had been arrived at through powder diffraction. A little strangely, the authors are persisting with reporting their gas-loaded Rietveld models, despite these clearly now being incorrect, but this is adequately addressed in the cif files where it is noted clearly that the guest locations are not determined accurately.

Answers and comments to the reviewer's comments

We appreciate the editors and reviewers very much for their constructive comments and suggestions on this manuscript (NCOMMS-18-18061-A). We have carefully examined the reviewer's comments, and revise the manuscript. The details of our specific comments are as follows, where the answer/comments from us are given in blue:

Reviewer #1

The revised manuscript has largely improved the previous version, and the inclusion of the single-crystal data of the structures under different gas atmospheres shows the structural transformations without any doubt, which previously were not that unequivocal with only X-ray powder data. The manuscript is suitable for publication.

>> Thank you for favorable comments.

I have however one comment for the authors to check: Figure S7, which shows the X-ray powder data with different gases (experimental vs calculated from single crystal diffraction), seems to have the wrong experimental powder diffractogram for O₂ at 130 K. The blue diffractogram (obs.) seems too similar to the red diffractogram (obs), which corresponds to N₂ at 130K. This is more evident by comparing with the calculated diffractograms, focussing in the region 14-16 degrees.

>> Thank you for comments. Firstly, we confirmed that the data in Figure S7 is correct. Actually, the powder data mentioned above is the same one as shown in Fig. 2b, and similar pattern under the same condition was also observed in a synchrotron PXRD measurement (Fig. S17a).

Then, the difference between calculated and observed pattern under O₂ may indicate a little difference in the lattice constant for powder sample and single crystal. Nevertheless, N₂ and O₂-I phase are highly similar as evidenced by the single crystal structures (Fig 2c, Table S4, Fig. S9), and PXRD measurement (Fig. 2b, Fig. S17), and hence, the slight difference in the lattice constant does not affect overall story of our manuscript.

The description about the above difference was mentioned in the supplementary information as follows:

SI Page S23, Supplementary Note 6, line 10: The slight difference between calculated and observed pattern under O₂ at 130 K indicate a little difference in the lattice constant for powder sample and single crystal.

Reviewer #2

The revised manuscript has successfully addressed all of the I raised in my previous review and in my view the manuscript is now suitable for publication – indeed, I think it will be an excellent addition to the literature. Most importantly, the authors provide single crystal X-ray structural determinations of 3 of the 4 gas-adsorbed phases, which has provided an enormous improvement in the structural understandings; notably, the concerns I had raised previously regarding the unphysical location of the gas molecules within the pores appear to have been justified, with the new single crystal determinations providing an entirely different picture of the host-guest configurations than those that had been arrived at through powder diffraction.

>> Thank you for a favorable comment.

A little strangely, the authors are persisting with reporting their gas-loaded Rietveld models, despite these clearly now being incorrect, but this is adequately addressed in the cif files where it is noted clearly that the guest locations are not determined accurately.

>> Thank you for a comment. We reported the models including gases, because the parameters (Table S7) and PXRD patterns (Figure S8) were actually calculated for the model with gases. However, as mentioned in your comment, we did not discuss the guest locations for them.

This point is already described in main text as follows:

Page 7, line 16: (the O₂ molecules for **1D-O₂-II** have less accuracy, so only the framework structure is discussed)

Additionally, we added the description about adsorbed gases in the reported structure of O₂-II phase in the supplementary information as follows:

SI Page S23, Supplementary Note 6, line 2: For the construction of the plausible initial structural model of **1D-O₂-II** that satisfies the observed PXRD pattern by a direct-space method, the introduction of gas molecules into the frameworks is absolutely imperative, although the assigned numbers and positions of gas molecules may not be entirely accurate. And thus, only the framework structure is discussed for **1D-O₂-II** although adsorbed O₂ molecules are reported in CIF file.